# Learning to Teach as a Spectator or a Participant—Ideas of Vocational Learning in Policy on Teacher Education

**Sandra Jederud**

School of Education, Culture and Communication, Mälardalen University, 722 20 Vasteras, Sweden; sandra.jederud@mdu.se

**Abstract:** This article explores how ideas about teacher student learning are expressed in policy documents, especially the school-based part of teacher education in the Swedish educational system. The study concerns how these parts of the education process are outlined through two reforms, one in 2001 and one in 2011. Of particular interest in the article is how these reforms represent, and to some extent, produce ideas for learning the teaching profession in relation to the specific context of practicum, and that this context pertains to specific knowledge, theoretical and practical knowledge. However, the analysis of the documents points towards a tension between theory and practice. By additionally analyzing policy documents on a local level, where the national policy is operationalized, the tension between theory and practice becomes more explicit. By introducing the notion of spectatorship knowledge and participatory knowledge, two different visions of students' learning emerge in policy. However, the relation between these two perspectives is problematic, as will be shown in the concluding part of the article. In conclusion, it is suggested that rather than muddling spectatorship knowledge and participatory knowledge together, teacher education programmes should provide for conditions where these knowledge forms are better taken care of.

**Keywords:** teacher education; school-based education; policy documents; participatory perspective; spectator perspective

## 1. Introduction

This article concerns ideas, expressed in policy, about how teacher students are supposed to learn the profession in school-based education (hereafter SBE) within Swedish Initial Teacher Education. Ideas about what constitutes adequate learning in teacher education (from now on, TE), and how this should be organized in order to fully prepare and equip student teachers (hereafter students) for their future profession, varies over time. The common denominator in these ideas is the tangible tension between theory and practice. The focus of the underlying study is how policy approaches this tension. Internationally, over the past two to three decades, a practice turn has been identified where new emphasis is on student teachers experiences in school and on inaugarating closer ties between universities and partner or practice schools [1]. Additionally, in Sweden, the significance of SBE within teacher education has changed over the last years. This is evidenced by the emphasis in policy concerning the introduction of specific practice schools in TE [2], as well as by the fact that eleven Swedish universities have been allocated funds in order to develop and conduct work-integrated teacher education programmes [3]. These changes point towards the increased political value given to learning through practice within teacher education programmes. As practice has gained higher importance in national policy, Swedish universities have interpreted these new ideas when organizing their teacher programmes and formed their own local policies that interpret the value of practice. Of special interest, then, is what happens in this process when the ideas reflected in national policy about learning the profession are transformed into a local policy context. Thus, due to the above new identifications of a practice turn within teacher education, both

internationally as well as nationally, during the last decades, this study concerns the two most recent teacher education reforms in Sweden (2001 and 2011) [4,5].

### 1.1. Aim of the Study

In relation to the above introduction, the aim of this article is to study how learning the profession of teaching within SBE in Sweden is outlined through national policies and how this is interpreted in policy at a local university following the teacher education reforms of 2001 and 2011.

To fulfill the aim, the research questions are:

1.　　How are ideas on vocational learning in SBE described in national policy?
2.　　Do the above ideas change over time, and if so, how?
3.　　How are these ideas configured in policy at a local institution of teacher education?

### 1.2. The Theory and Practice Tension in Teacher Education

The conditions that outline teacher educators' professional actions are highly related to the ways that the discourse of theory and practice contributes to the perception of vocational or academic focus within teacher education [6]. It is in the interest of political governance to repeatedly return to the questions of how theory should be connected to practice, and visa versa. However, the question is if it is reasonable to expect that these boundary problems are solvable, and if so, what are the means that are available in relation to what should be submitted to teacher education to handle professionally [7] (48–54). Policy-makers have constantly had opinions regarding SBE within teacher education, and this study concerns how new directions of policy are configured within teacher education and how they are implemented on a local policy level.

TE can be considered a professional education, and even if there are differences in how such a vocational education is organized, there are specific features that characterize it—these entail theoretical as well as practical elements. Additionally characteristic for professional education is that there is a tension between these two elements [8,9]. Agewall and Olofsson [8] describe that there is a difference between how theoretical and practical knowledge is organized within different educations and how they therefore characterize the structure of the education. This means that they are, to various degrees, established either theoretically or vocationally. TE has gone from a strong vocational establishment to a more scientifically theoretical education, which can affect how newly educated teachers perceive that they are prepared for their coming profession.

Theory and practice are both essential within TE, but are, according to Schulz [10], a double-edged sword, where an over-emphasis on only one aspect will diminish its function. Additionally, Damgaard Knudsen and Haastrup [11] show that theory and practice are categories of knowledge that can be perceived in different ways, and that there is a variety of theory and practice relations. A point of departure is to adress the issue as 'gaps' rather than a 'gap' between different theory and practice understandings. Work on these different relationships between the two contexts can be understood as a procedure of awareness which seeks to illuminate what theory and practice is being implemented and whether this is suitable [11]. These gaps were one of the major drivers for a new teacher education reform in Sweden 2001 [4], as "In the interaction between proven experience and scientifically grounded knowledge, thoughts as well as action can be developed so that practical knowledge emerges. Practice and theory must therefore be brought closer together by connecting theoretical studies to pedagogical activities" [4]. Coming to the next teacher education reform in Sweden 2011 [5], the above dilemma was still current, and one of the drivers was, therefore, to extract practice as a part of other educational areas and create a separate element to emphasize the importance of this part of teacher education.

In which settings should the learning of practical and theoretical teaching skills take place, and to what extent, is broadly debated and can differ greatly between countries. In England, for example, teaching is described as a skill that evolves in practice (here implying that TE can be moved from universities to practice schools), whereas in Scotland,

the discussion characterizes teaching mainly as a profession that calls for university-level TE where practicum is not emphasized [12]. The organization of Norwegian TE programs is closely connected to the domain of professional practice, and is, in fact, conducted through learning-by-doing. The 2006 reform of Norwegian TE strongly promoted connections between the learning outcomes of practical training and topics and theories in the curricula at the university colleges. This is due to evidence of a weak agreement between what is emphasized in practice and current theoretical studies [13]. Finnish TE programs have, as a principle, to initiate teaching practice at an early stage of TE and to incorporate theoretical aspects. A number of practice periods follow each other, at either specific practice schools or regular schools, the main idea being to implement a form of practice permeated by theoretical aspects, as a research-based approach is the key link in the process of teaching [14].

Teacher educators worldwide have engaged in multiple efforts to create a more hands-on and "practice-based" TE (see Forzani [6] (p. 354), for an account of these efforts in the USA), and initiatives such as establishing partnerships with specific teacher training schools have been undertaken in Finland [15] as well as in the USA, where, however, they go by the name of professional development schools [16]. In contrast to this, numerous teacher educators have also discussed the idea that establishing a tighter connection to practice should not be limited to specific training schools. Coursework and skills developed at the university are equally important arenas for aligning professional learning with practice [17].

Ahlström and Kallas [18] offer the criticism that TE in Sweden seldom makes use of evaluations for change or development. Instead, changes in TE take place due to experiences from schools or knowledge that teacher educators have from their own experiences as teachers. Additionally, changes such as prolonging subject studies or SBE within TE are implemented without further analysis of what type of knowledge a teacher actually needs in order to function within the profession [18]. Today, TE in Sweden is criticized for the fact that education is mainly about teaching, and only a small part is "hands-on" and provides possibilities to meet the profession in SBE [19].

Further, the Swedish political discussion concerning TE is surrounded by disappointments, a perception of inadequacy and incompleteness regarding a school that does not meet measures, and of a teacher education that does not focus on the most important teacher skills. The teacher education reforms in 2001 [4] and 2011 [5] are pervaded by these disappointments. In 2001 [4], a need was seen for a teacher education that could provide adequate subject knowledge as well as pedagogical competences. However, "how the weighting of different educational goals within teacher education should take place is not obvious" (p. 417). In 2011 [5], the foundation of the new reform was grounded in evaluations of criticism towards a teacher education lacking a scientific foundation, where students are perceived to have too much freedom of choice and where important areas of knowledge are lacking. Furthermore, other governmental measures, such as further emphasizing the importance of SBE through organizing work-integrated teacher education programmes have taken place. An order was given by Government Offices [20] to, by this action, increase the number of applicants and further, to strengthen teacher education.

Regardless of how these disappointments are manifested and how they are related to a specific view of knowledge, it also gives expression for experience-based understandings of the main role of TE [21]. Following, these disappointments lead up to drivers and barriers for designing TE, which is seen in numerous attempts to readjust SBE in Swedish teacher-training programs [2,4,5] Today, this part of TE recurs over a period of 4–5 years, and the way it is organized can vary at different universities. Usually, a period of 20 weeks in total is spent in a school context in order for students to obtain teaching skills by taking part in teachers work under guidance from a supervisor.

*1.3. Learning in SBE through a Spectator Perspective or a Participant Perspective*

When organizing practice within TE, indications are put forward as to how the profession should be learned rather than what should be learned. Learning entails obtaining

knowledge and looking back at Aristotle's categories of knowledge as they are described in Nicomanean Ethics; there is a qualitative difference between theoretical and practical knowledge. One of the main ideas behind these categories, according to Saugstad [22], is that their nature and characteristic features are closely connected to their aim and the function. Theoretical knowledge can metaphorically be described as "spectator knowledge" that can be learned through a spectator perspective (from now on, SP) where the aim is to understand and to explain [22]. Here, an essential part of learning the vocation through SBE is to be able to take a step away from the social context as to enable seeing things from a distance, in order to recognize underlying values and ruling patterns. The role of the supervisor is to be a critical interlocutor, but also to support the student when problematizing and reflecting the profession of teaching. Practising the profession here means training the ability to question one's actions and weigh different alternatives against each other [23]. In this theoretical position, it is vital that there is a variation within the periods of SBE, that there is a possibility to obtain experience from different organisations, and that the periods of SBE are kept short. Practical knowledge, however, is participatory knowledge that can be learned through a participatory perspective (from now on, PP) where the aim is to be a part of a social context and learn to act accordingly [22,24].

Acquiring practical knowledge in SBE is dependent on a context, is subject for change, and is unpredictable, as it is linked to concrete and specific situations. The role of the supervisor here is to supply valid knowledge, spoken as well as unspoken, and the main task is to be a role model and bring forth tasks that are sufficiently difficult whilst the student grows in to the profession of teaching. Learning is seen as a social process, a process that takes time and begins in a peripheral position as an apprentice, then advances to a central position as a master, being a central part of the professional community [24]. Here, learning is based on doing, and knowledge grows out of experience and can only be acquired by participating in the actual situation, as opposed to a SP where learning is based on vocalized, distanced and critically theoretical reflections [23]. Participant knowledge can also be defined tacit knowledge that is implicit in our models of actions; a spontaneous performance that is difficult to explain in words, according to Kinsella [25]. Instead, it is transmitted by showing and is labeled basically as know-how. Tacit knowledge is transferred by physical traditions and actions [26].

These two different ways of learning are equivalent to epistemological positions, where, on one hand, learning through a PP facilitates the positioning of the student in the very centre of teaching practice, and this procedural knowledge is attained when the student has the ability to cope with specific circumstances. On the other hand, learning through a SP is decontextual and positions the students in a more peripheral position, where they are instead facilitated to create propositional knowledge, which entails a distanced and critical view of what is going on. Hegender [27] expresses that a student teacher can learn to express propositional knowledge about a particular situation, but this does not mean that he/she has the procedural knowledge to deal with it, or vice versa.

## 2. Materials and Methods

This is a study examining policy regarding the two most recent teacher education reforms in Sweden (2001 and 2011) [4,5]. The method used is a qualitative content analysis that extends to sorting large amounts of text into organized categories that signify similar implications [28]. The aim of this content analysis is "to provide knowledge and understanding of the phenomenon under study" [29] (p. 314). In the present study, the analysis is concentrated on a critical seeking of content and meaning rather than on deriving or analyzing policy as the articulation of a specific discourse or an analysis of what kind of authenticity and subjects different discourses form [30]. The analysis is conducted in relation to how ideas on learning in SBE, connected to the spectator or the participant perspectives, are visualized in national policy documents and further interpreted in local policy documents. As such, it is influenced by Ball's [31] notion of context of influence, meaning that any interpretation of policy should stretch out beyond the idea of conceptual-

izing policy as a neutral text that is passed down in order to establish change. According to Ball, new policies do not normally outline what to do, rather, they create circumstances in which the amount of accessible options are restricted or altered, or specific goals or possibilities are established.

### 2.1. The Documents

Policy documents concerning the curriculum of teacher education can be visualized and applied on different levels. Van der Akker [32] shows of five different levels, from "Supra" on an international level to "Nano" on an individual level and also gives examples of how the curriculum is used in the different levels.

The empirical material in this study, that is governmental policy documents as well as local policy documents, would, according to this categorization, be categorized as "Makro material, that is policy documents on a national level and "Meso material", that is policy documents on an institutional level (p. 2). According to Van der Akker [32], each level is affected by the levels that are above it in the schedule, as they draw the terms for what happens on the lower level. However, according to Taba [33] it can not be disregarded that influences also go in the other direction; that the higher level adapts to what it intends to affect, as the lower levels draw the guidelines for what is possible regarding formulation and realization on the higher levels. In order to best be able to ascertain how changes in practice within TE are implied by certain expressions in government bills and the stated reasons for changes on a local level, the below policy content analysis was conducted.

The above documents were chosen because each can be seen as a crucial part of the argumentative process that has led up to changes in TE on a local level. SOU (1999:63) [4] and SOU (2008:109) [5] are government commissions of inquiry consisting of parliamentary committees appointed by the Ministry of Education. These committees submitted proposals for new TE directives, suggesting necessary changes which are then developed into two Propositions (1999/2000:135; and 2009/10:89) [34,35]. These are government bills that were sent to parliament with suggestions for reforms of TE, and which led to the reforms of 1999 and 2009. Additionally, in connection to the above, local course syllabuses (2007) [36] and local program curriculums (2012; 2014) [37,38] at one university that provides teacher education in Sweden were analyzed. Finally, (Promemoria U2013/4305/S, 2013) [2] is an invitation to universities to participate in a pilot project comprising specific practice schools, and SFS (2014:2) [39] is a governmental regulation regarding this pilot project. Accordingly, a local SBE handbook [40] (2013) and local application [41] (2014) to take part in the project were examined (see Table 1).

**Table 1.** Analysed policy documents on makro and meso level.

| Policy Level | Local Level |
| --- | --- |
| SOU 1999:63 | Local course syllabus 2007 |
| Prop. 1999/2000:135 | Local program curriculum 2012 |
| SOU 2008:109 | Local program curriculum 2014 |
| Prop. 2009/10:89 | Local SBE handbook 2013 |
| Prom U2013 | Local application pilot project specific |
| SFS 2014:2 | practice schools 2014a |

### 2.2. Analysis of Documents

The practical approach to the data material used in the study is grounded in Säfström and Östman´s [42] purpose-related text analysis, the aim of which is to read texts in an exploratory manner, and also to study how different writings clarify different perspectives. Additionally, a concept of ideal types was used, which is a form of idea construction. These ideal types act as analytics, being able to refine certain traits and act as patterns to overlay texts. For example, we can, according to Bergström and Boréus [43], choose ideas that are relevant for the matter, and here, emphasis is on how certain expressions in reports from government commissions of inquiry and government bills give attention to ideas

about how to learn the profession of teaching in SBE. I have focused on the explicit content and meaning, that is, what is expressed in words in the text. To begin with, the reading concerned the above documents and was focused on the specific parts where SBE was mentioned in the texts. As ideal types, guiding the readings and following analysis, we have used Saugstad´s [22] categories of knowledge in order to specifically find and analyse parts where different perspectives of learning are at stake. In the final discussion, we have made more explicit connections to the two perspectives of learning, PP and SP, using the notion of participatory conditions and spectator conditions. This is in order to suggest, in line with Saugstad [22], that in order to keep the boundaries between spectator and participatory knowledge, one needs to give careful consideration to the organizational contexts and prerequisites of TE.

## 3. Results

This results section is structured according to the research questions of the study. The first paragraph answers the first two questions and the second paragraph answers the third question. The excerpts are chosen due to their function as examples in the context of the study. They depict relevant policy changes and/or work as examples of articulations or intentions concerning how teacher students are to be provided with opportunities for learning within TE.

### 3.1. How Ideas on Vocational Learning in SBE Are Described in Policy and How These Ideas Change over Time

Every reform of the Swedish educational system has brought forward a specific mixture of basic ingredients within SBE. The government public commission *To Teach and Lead—Teacher Education for Cooperation and Development* [34] suggests the following concerning the structure of SBE within TE (this and following citations are the author's translation):

> SBE within teacher education should prominently comprise practical exercises and exposing the students to a smaller amount of teachers´ teaching styles, is not adequate. [. . . . ] Practice needs to consist of more real-life experiences. Students need to take part in lifelike and realistic exercises within their education as to able to "construct" their own solutions and reflect upon these. [34] (p. 100)

In this text, there is an evident critique of the current organization of TE and its inability to provide teacher students with real life experiences. It outlines the limitations of having students encounter experience from a limited number of situations and teaching styles. In the example, the need for more realistic participation in authentic experiences is shown. This shift from doing exercises to performing actual teacher work is further emphasized by the government public commission [34], in which it is stated that schools accepting students should now be named "partner schools", not only to build up a partnership between these schools and teacher educators, but also to tie a group of students to the one and the same SBE school.

> The ambition is that students shall follow development during a longer period of time and get the opportunity to enhance their understanding for the local conditions of the pedagogical organization. By this concentration, students are expected to familiarize themselves to the culture and to learn the school code that is specific for the practice/partner school. [34] (p. 103)

Yet, even though students should be at the same partner school, being connected to a teacher team is emphasized rather than being connected to the same supervisor and same classes:

> During SBE, the student should plan and conduct pedagogical activities in varied group sizes, actively follow the planning of the teacher team and participate in the ongoing work that the teacher team is responsible for—all together with one or several supervisors from the teacher team. [34] (p. 104)

The same public commission further submits:

> At the preschool, school, recreational center and adult education center, students are expected to cooperate and teach the same children, adolescents or adults. [...] on the basis of their diverse experiences and perspectives, students should discuss questions that are of importance for all teachers so that they can develop a shared perspective on child, adolescent and adult learning and socialization. [34] (p. 127)

Ideas about how to learn the profession of teaching are expressed here as a desire for students to be a part of a teacher team, but also that they should have the opportunities to take a step away from the organization and reflect together with others, implementing practice for all age levels. In the following excerpt, one can see how a more realistic practicum is considered to provide qualitative reflections amongst students with other practicum experiences:

> The school based part of the general field of education enfolds possibilities to learn while working and gives the students opportunities to develop their professional skills as well as their abilities to reflect. During the school based part, knowledge such as discussions with children, pupils and parents and work in teacher teams should be included. Such knowledge is obtained through experiences from many different situations. [4] (p. 19)

As part of the teacher education reform *A Sustainable Teacher Education* [35] a stronger emphasis on students dealing with a variety of tasks during practice was suggested.

> Additionally, it is appropriate to schedule practice periods at different times during the school year to enable insight into various special tasks that teachers deal with, for example, grading or long-term planning. It is also of importance that practice comprises examples of other recurring tasks that are part of the teacher profession such as parent meetings and local development work. This also places demands on students' engagement and flexibility. [35] (p. 399)

This envisions a focus on school work as a holistic endeavor, being comprised of many different tasks and responsibilities. In the draft, this holistic approach is seen in the emphasis put on the students being part of a rich context and learning within this context.

A significant change in policy is that SBE should now not take place at "partner schools" any more, but at "field schools". Instead of SBE coordinators having to convince schools to accept students, schools are now instructed to apply to be accepted as "field schools". After a quality review (by the Swedish National Agency for Education/National School Inspection), the most appropriate schools are selected. This will give the "field school" a seal of quality, which is of great value in the competitive situation in which schools of today find themselves in. Furthermore, SBE should now be organized as a course of its own within teacher education programmes and consist of 30 ECTS [5].The fragmentation of supervision in a teacher team that was emphasized in SOU 1999:63 [34] is now seen as problematic in relation to responsibilities and examinations, as well as to the character of career steps for the individual supervisor. Instead, in SOU 2008:109 [35], it is accentuated that:

> Only permanent employees and qualified teachers who have at least three years experience can be appointed supervisors. The teacher should also have completed adequate further supervisor education at an institute of higher education. Guidelines for reasonable compensation for supervisors should be established by the committee. The position of a supervisor is a very important assignment that should be seen as a career step for a teacher. [35] (p. 403)

Here, the importance of the supervisor is highlighted, however it is not specified what the further supervisor education should entail regarding content and credits, only that it should take place. Further is expressed concerning SBE:

> Having teacher practicum in the local area can be of importance for the student for social reasons but also as a way to strengthen the connection between TE

and schools. Universities should therefore primarily provide teacher practice at schools in the immediate geographic area. [35] (p. 400)

Here, the social and contextual connections between TE and the schools and student teachers and their field schools is envisioned. In the following excerpt, considerations regarding having practicum too early are being raised.

> . . . . . . the investigation recommends that SBE is given in a smaller amount of longer periods. For an education of four years, it can for example be appropriate to place SBE in for periods of about a month each. It is also sensible that the first period is not placed too soon. [ . . . ] Having SBE too soon enfolds a risk of becoming pure auscultation. [35] (p. 399)

The above excerpt signals that, even if practicum is seen as essential for learning the profession, it should not take place too early in the programme. This, since it would lead to pure auscultation. It thus is assumed that practicum must be based on a sufficient knowledge base in order for teacher students to practice teaching, but should be led by knowledge and reflection. In the governmental bill from 2009, (Prop 2009/10:89) [5], it is advised that practice should encompass 30 ECTS and take place within relevant age categories and school forms, as well as within relevant subjects or subject areas, so as for SBE to prepare students as much as possible for their future profession.

Coming to 2013, a third renaming of SBE schools is implemented when "field schools" were instructed to become "specific practice schools". Additional action concerning SBE is taken in Prom U2013/4305/S [2], where it is suggested that a five-year trial period with specifically targeted government funding shall be distributed to participating universities that organize teacher practicum at specific practice schools [2] (p. 4):

> A pilot project with specific practice schools and specific practice preschools should be characterized by a high concentration of students and further educated supervisors. SBE should be a natural part of the organisation and supervisors should be highly competent so that the students´ development can be furthered. By having a high concentration of students and supervisors, there should be adequate possibilities for exchanging experiences both between supervisors and students as well as amongst themselves [ . . . ] All supervisors should have 7.5 ECTS in leadership training, or the equivalent. [2] (p. 4)

Here, importance is given to a high concentration of both students as supervisors, and the supervisors are to have acquired special competence through further education. The excerpt states the importance of students being given the opportunity to be in the midst of vocational practice for longer periods of time, but also that the context for practice contains supervisors with specific competence for providing the students both practice and exchange of experiences. The importance of supervisor competence is further strengthened through the requirement of attending additional courses of 7.5 ECTS in leadership training. Therefore, there is an indication of that it is not sufficient for students to learn in any practical context, as this context should provide supervisors with specific knowledge and roles.

In 2014, a governing ordinance named "A Pilot Project with Specific Practice Schools and Specific Practice Preschools within Teacher and Primary Teacher Education Programs" (SFS 2014:2) [42] was given. Fifteen universities and university colleges applied for and were accepted to the project in order to arrange their teacher practice at a limited number of specific practice schools. In the ordinance, it was stated that students were to implement "SBE at specific practice schools with different prerequisites" [39]. The ordinance expresses a vision of students being at specific practice schools throughout their teacher education, indicating the importance of gaining specific context knowledge in order to learn the profession through more professional preconditions. By remaining in the same communities of practice, students can become a part of this context and learn to act accordingly, this, of course, depends on how local universities interpret national policy. However, the ordinance also states that students are to be given the possibility to gain from schools with different

prerequisites. Whether these experiences are to be acquired at the same specific practice school or different specific practice schools is not clarified.

*3.2. Changes on a Local Level and How Policy Ideas Are Configured in Policy at a Local Institution of Teacher Education*

Following TE 2001, as mentioned above, various measures were taken to change SBE within TE. The aim was to enhance cooperation and also to provide equal conditions for preschool teachers and schoolteachers as a way to establish similar professional identities. Students should, within SBE, reflect on questions that are of importance for teachers within all age categories. This was interpreted in the following manner by a course syllabus—practicum in a local TE 2007 [36]:

> Teacher practice is a part of all three blocks of course studies and comprises pedagogical work, observations and interviews in various learning environments, listening, reflecting, and teaching exercises, as well as research and development work. [36]

In the course syllabus is stressed the importance of seeing teacher practice as a complex endeavor, consisting of different competences and knowledge types. The students should thus be provided with opportunities to try out different vocational situations as well as more research like situations in order to create a foundation for reflection and further learning. Exactly what the expression of pedagogical work should include was not specified, and therefore, it could be interpreted by each partner school individually.

The excerpt from the syllabus above shows signs of seeking a more general-knowledge type of teacher, who not only has experience from different learning environments and levels, but also can more easily be introduced into the workforce. National guidelines emphasize the unity of theory and practice, as well as cooperation between student teachers and teachers within various school forms. At the local TE program of the study, this is transformed into an emphasis on partner schools where students should cooperate continuously with a team of teachers in order to gain insight into a broad spectrum of school activities.

Following TE 2011 [35], there was a significant change of focus in the program curriculum at the local TE (2014) [38]. Now, emphasis is on relevant subjects and school forms in accordance with the chosen TE program. While it was previously emphasized that students should conduct research work such as observations, interviews and questionnaires in connection with lessons, and listen and reflect over teaching exercises, these activities have now been downplayed.

> Practice shall be located to relevant ages and subject areas and is to comprise the following: planning and performing pedagogical activities with personal responsibility under supervision, listening, observing and reflecting in connection to everyday school activity, participation in general competence development at the assigned practice school. [38]

The above excerpt shows how the main focus of practice should be on actual doing and experiences that can only be attained in situations within the specific context of the assigned practice school. Although observations and reflections are seen as important, it is the possibility of gaining actual teaching experiences that stands out as most important.

The transformation from national to local policy can be traced though an application to take part in the national project regarding specific practice schools [39].

> SBE should be a central and important part of the organisation at the specific practice school. The aim is to create a high concentration of student teachers as well as qualified teachers who are further educated within supervision. The amount of supervisors at every specific practice should be at least six. Every supervisor should be able to supervise two students at the same time which facilitates cooperation between the students. [39]

The above is further specified in a handbook for SBE [40] at a local university as follows:

> Being a specific practice preschool/specific practice school comprises that students and supervisors are concentrated at a smaller amount of schools. The supervisors at the specific practice preschool/school form a team of supervisors that are joint responsible for the students placed there. The team of supervisors also contribute to the development of SBE within the specific practice preschool/school. The concentration of students within the specific practice preschool/school facilitate for peer learning between the students during the courses of SBE. [40]

In the application, an image of close collaboration between the university and the practice schools is depicted. First, the student is placed at a specific practice school throughout the entirety of their studies. Second, the role of the supervisor is emphasized as an even more important figure who should contribute to the development of SBE within the specific practice school. Third, besides having the same specific practice school throughout their entire teacher education, they are to do what is described as field studies:

> The student should do shorter, temporary excursions during their education. The temporary excursions are done as to provide for as much experience as possible from the SBE courses and for the student to experience the diversity that he/she will meet in his/her occupation [ ... ] The length of the excursions can vary depending on the aim of the excursion.

The content of the excerpt shows signs of providing a counterweight to the practical nature of the practicum by having students conduct small excursions and field work, where they can work on the more research-based underpinnings of TE.

## 4. Discussion

The aim of this article has been to study how learning the profession of teaching within SBE is outlined through national policies and how this is interpreted at a local university following the teacher education reforms of 2001 and 2011. In Sweden, as well as across Europe, there has been a trend of increasing the amount of school-based practice within TE programmes [2]. The most recent teacher reform "A Sustainable Teacher Education" in Sweden stated that "teacher education should provide becoming teachers with a solid base of knowledge and efficient tools to be able to practice the profession in a professional and secure manner" [35] (p. 20). Although this can be assumed to be a prevailing vision through all TE reforms, there still remain uncertainties regarding how SBE within TE is to facilitate the above to take place. When it comes to the structure of SBE, an ongoing balancing act can be seen, from promoting learning from a SP to learning from a PP. There is clearly an ongoing back-and-forth between learning through these two perspectives that tends to move from one side or the other, and results from the study indicate that emphasis has gone from ideas about mainly learning from a SP to ideas about mainly learning from a PP.

### 4.1. The Tug-of-War between Theory and Practice

Pervading through the documents analyzed in this study is the idea that the ongoing tug-of-war between the two perspectives on learning: participatory learning and spectatorship learning, indicates that there is an ambivalence to what should be given most value when students are learning the profession. When it comes to significant changes throughout this period of time, the renaming of SBE taking place in "partner schools" to "field schools" and further to "specific practice schools" can be related to the value that is given to either perspective. Emphasis on SBE within "partner schools" [34] pays attention to a shared responsibility, not only between higher teacher education and schools accepting students, but also for supervising students who were placed in teacher teams with shared responsibility. SBE was part of other courses and could be shared up to a number of periods.

Agewall and Olofsson [9] have described that TE has gone from a strong vocational establishment to a more theoretical education, which can affect how newly educated teachers perceive themselves to be equipped for their coming profession. However, this study shows that the balance now seems to be tipping in the other direction, that SBE is given higher value and that students are to spend longer periods of time at specific practice schools with further educated supervisors. The renaming to "field schools" [35] had significance in the meaning that not only did schools have to apply to accept students and become a "field school", which was meant to raise the status of the engagement, but also, "field schools" accepted students for longer periods of time. SBE became a course of its own within TE, as to guarantee students a minimum length in SBE activities, and an emphasis was put on that supervisors should be highly skilled and experienced. Additionally, field schools received a higher amount of compensation than earlier for accepting students. SBE is given a higher value in many aspects and the balance is now instead tipping to students learning through a PP, as they are still to be at the same "field school" in the periods of SBE, and they are to have individual, skilled supervisors, instead of belonging to a team of teachers.

Accordingly, when "field schools" are renamed "specific practice schools", these are still to apply to accept students and targeted government funding shall be distributed to participating institutions that organize teacher practicum at specific practice schools [2]. Here, learning through a PP is mainly given value, where it is important that the student is given the opportunity to be in the midst of vocational practice for longer periods of time, where expertise and knowledge is obtained. As described by Larsson [44], this means that having SBE at one specific practice school accentuates learning through an apprenticeship where the supervisor is in focus. Having SBE by one specific practice school accentuates learning through an apprenticeship where the supervisor is in focus. According to Lindqvist & Nordänger [23], the supervisor plays the important roles of being a role model and of bringing forward tasks on an adequate level as the student grows into the profession.

### 4.2. Separating or Combining Theory and Practice? Going from National to Local Policy

Following Saugstad, theoretical and practical knowledge are two different types of knowledge that have different characteristic features and demand different procedural learning. In order to provide for these two and to facilitate effective support for students to learn, "a systematic organization for, and a coordination of the different processes of learning need to be established" [23] (p. 5). This systematic organization seems problematic, as the analysis of the documents shows that there are evidently difficulties in facilitating SBE regarding how the profession of teaching is best learned. Evidently, something happens when national policy is interpreted to local policy, as organizational problems arise. When it comes to the pilot project "A Pilot Project with Specific Practice Schools and Specific Practice Preschools within Teacher and Primary Teacher Education Programs" [45], it is stated that students are to be given the possibilities to implement SBE at specific practice schools with different prerequisites, which indicates an ambivalence in itself. First is a vision that students should be placed at the same specific practice school throughout their teacher education, which implies an importance of learning the profession of teaching through a participatory perspective, as students remain in the same community of practice and become a part of this context. However, at the same time, students are to be given the possibility to gain the experience of different prerequisites. If this is to be acquired at the same specific practice school or different specific practice schools is not clarified. Nevertheless, it does designate that they, at the same time, should be given the possibility to experience different organisations, limiting their time for developing participant knowledge in a familiar context, such as at the practice school. At the local TE, this is solved in a way that enables students to do shorter, temporary field studies during their education. These temporary excursions can, however, vary between one hour to two days during SBE 1 and SBE 2, and one hour and two weeks during SBE 3. Here, the question remains as to what

shorter, temporary excursions entail, and if a minimum of three hours of SBE experience of different prerequisites, for example, with another supervisor at the same school, can be claimed to meet the intentions on a national policy level.

## 5. Conclusions

When organizing TE, one must consider practical, political and pedagogical factors. The political dimensions are seen in national steering documents that contain several ideas that do not always harmonize. This has to do with the fact that there are different political ideologies that affect the content of the steering documents, as well as schools themselves. Furthermore, when interpreting and implementing national policy, one needs to consider the pedagogical challenges of making possible everything that is asked for in the steering documents, as well as the practical conditions affecting the possibilities for fulfilling the growing expectations of practice within TE. Reforms are, according to Ball [31], not merely about altering the way matters are organized or performed, they are about rethinking education and what it actually takes to be educated. This study shows how policy reforms within Swedish TE and a local policy reform have led to a problematic mix between spectatorship learning conditions and participatory learning conditions. The local university involved in the pilot project with specific practice schools and specific practice preschools within teacher and primary teacher education programs [45] is faced with a dilemma, as they are required to have a limited amount of practice schools, and therefore, do not have the means to facilitate students' need to implement SBE at specific practice schools with different prerequisites. Furthermore, policy on a national level has, through the pilot project, made an attempt to restore the balance between the two different types of knowledge: participatory knowledge/learning and spectatorship knowledge/learning, both of which demand different learning opportunities.

However, this national policy attempt, when interpreted on a local policy level, encompasses organizational problems in order to coordinate the establishment of these different opportunities. This is in accordance with Ball [46], who states that new policies normally do not draw out the framework to what is to be done, but instead manufacture circumstances where the extent of options are limited or possibilities are determined. There seems to be a will to facilitate both to take place, however this ambivalence is problematic, as local institutions are faced with organizational and pedagogical difficulties. One possible solution to this dilemma could be to better articulate and construct spectatorship conditions, as well as participatory conditions. This solution, however, would call for a strengthened collaboration between universities and practice schools.

**Funding:** This research received no external funding.

**Informed Consent Statement:** Not applicable.

**Data Availability Statement:** Not applicable.

**Conflicts of Interest:** The author declares no conflict of interest.

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
