# Peer review of "Learning to Teach as a Spectator or a Participant—Ideas of Vocational Learning in Policy on Teacher Education"

_education, doi:10.3390/educsci12100726_

Round 1

Reviewer 1 Report

The article is focused on a relevant topic and it is presented in a well organised way. The theoretical framework is adequate but not very recent. The methodological design is adequate to the research questions and it was well implemented.

The results and the conclusions are interesting and relevant both at a local and an international level.

The final references must be presented according to the MDPI rules.

Author Response

The theoretical framework has been updated with relevant references of value for the theoretical reasoning. 

The final references are presented according to MDPI rules. 

Reviewer 2 Report

Thank you for this interesting research article exploring the development of the Swedish teacher education system with reference to the theory vs practice debate. The tensions here are clearly laid out, however for greater clarity I recommend:

1 some brief contextual details concerning national TE policy in 2001 and 2011. 

2 the drivers for change in 2001 and 2011

3 a rationale for the study now  in 2022  I.e. how relevant is it to the current policy context.

Please also include an ethical declaration for the use of the author’s home institution as part of the pilot group of participating universities and some background on the number of trainee students enrolled each year and the length of their training - 4 years.

it might be useful to consider next steps for the research e.g canvassing student teacher views or participating schools in the programme as a means of exploring the organisational and theoretical issues raised here.

Author Response

Regarding point 1, some brief contextual details concerning national TE policy in 2001 and 2011 are clarified on page 4. 

Regarding point 2, The drivers for change in the 2001 and 2011 are clarified on page 3.

Regarding  point 3, a rationale for the study now  in 2022  I.e. how relevant is it to the current policy context, this is clarified on page 4. 

Reviewer 3 Report

A brief summary

This is an interesting paper with a clear and straightforward design, which makes a contribution to the growing collaboration between universities and practice schools.

The aim of this article was to study how learning the profession of teaching within SBE is outlined through national policies and how this is interpreted at a local university following the teacher education reforms of 2001 and 2011.

Ø  A question for the authors/: Why this research is focused only in two reforms: 2001/2011?

Is neded a historical view (before 2001) and what is the situation now in 2022?

So in Sweeden since 2011, no other educational reform was outlined?

This should be clearly analysed.

Specific comments

Ø  In the paragraph (line 133 - 137) the author is cited twice. Please put quotes if there is a citation or mention only once the author if it is a paraphrase. The page is missing too.

One of the main ideas behind these categories (Saugstad, 2006) is that their 133 nature and characteristic features are closely connected to the aim and the function of 134 them. Theoretical knowledge can metaphorically be described as “spectator knowledge” 135 that can be learned through a spectator perspective (from now on SP) where the aim is to 136 understand and to explain (Saugstad, 2006).

Ø  Most of the cited papers miss the page number:

(Forzani, 2014 - line 52 / Krantz, 2009 - line 57 / Freidsson, 2001 - line 65 / Biesta, 2012- line 87 /Schulz, 2011 - line 73 / Kansanen, 2003 – line 97 …. etc. )

Ø  Some punctuations to be checked and minor spell mistakes.

The tittle: “Learning to teach as a spectator or a participant ideas of vocational learning in policy on teacher education”  

 should be

“Learning to teach as a spectator or a participant, ideas of vocational learning in policy on teacher education”

or

“Learning to teach as a spectator or a participant - ideas of vocational learning in policy on teacher education”

(line 243) – the word learing (learning )

(line 261 )– the word exercizes (exercises) and there is twice is.

(This shift, from doing exercizes to perform actual teacher work is is further)

(line 577 )– the word univerisities (universities).

Ø  Some references cited in the paper are not included in the Reference list

Ahlström and Kallos, 1996     - line 107

Hegender, 2010                       - line 166

Taba, 1962                              - line 196

Säfström 218 and Östman’s 1999 - line 218

Sund, 2014                              - line 221

Flick, 2011                              - line 227

In the paragraph (line 75-80) the author is cited twice. In the reference list is only Damgaard Knutsen and Haastrup (2016) - Correct the name Knudsen

 (Damgaard Knudsen & Haastrup, 2015) is not in the reference list, or is to be corrected the year.

Ø  In the Reference list

Line 620 - Akker, J. van den. (2003):     correct Van

Line 702 /706- Lindkvist, P. & Nordänger, U-K. Linnéuniversitet (2010).  correct               Lindqvist

Line 740 - Thorsen, K. P – add the year

Ø  If author/s find appropriate can also include these papers:

Daniel Sunberg (2021). Evidence in the History of School Reforms in Sweden, DOI: 10.1007/978-3-030-66629-3_6

Richards, Jack J., and Richard Schmidt (2002). Teacher education. Longman Dictionary of Language Teaching and Applied Linguistics 552: 554.

Author Response

The following measures have been taken:

Page numbers have been written in where missing at cited papers.

Title has been changed accordingly. 

Spelling mistakes have been addressed. 

Missing references have been written in reference list. Corrections have been made in the reference list.

The questions regarding Why this research is focused only in two reforms: 2001/2011, Is needed a historical view (before 2001) and what is the situation now in 2022 have been clarified on p 2-5.